# Quantifying within-city inequalities in child mortality across neighbourhoods in Accra, Ghana: a Bayesian spatial analysis

Honor Bixby [1,2] James E Bennett,[3,4] Ayaga A Bawah,[5] Raphael E Arku,[6] Samuel K Annim,[7,8] Jacqueline D Anum,[7] Samilia E Mintah,[7] Alexandra M Schmidt,[1] Charles Agyei-Asabere,[5] Brian E Robinson,[9] Alicia Cavanaugh,[9] Samuel Agyei-Mensah,[10] George Owusu,[11] Majid Ezzati,[3,4,5,12] Jill Baumgartner[1,2]

For numbered affiliations see end of article.

**Correspondence to**
Dr Honor Bixby;
honor.bixby@mcgill.ca

## ABSTRACT

**Objective** Countries in sub-Saharan Africa suffer the highest rates of child mortality worldwide. Urban areas tend to have lower mortality than rural areas, but these comparisons likely mask large within-city inequalities. We aimed to estimate rates of under-five mortality (U5M) at the neighbourhood level for Ghana's Greater Accra Metropolitan Area (GAMA) and measure the extent of intraurban inequalities.

**Methods** We accessed data on >700 000 women aged 25–49 years living in GAMA using the most recent Ghana census (2010). We summarised counts of child births and deaths by five-year age group of women and neighbourhood (n=406) and applied indirect demographic methods to convert the summaries to yearly probabilities of death before age five years. We fitted a Bayesian spatiotemporal model to the neighbourhood U5M probabilities to obtain estimates for the year 2010 and examined their correlations with indicators of neighbourhood living and socioeconomic conditions.

**Results** U5M varied almost five-fold across neighbourhoods in GAMA in 2010, ranging from 28 (95% credible interval (CrI) 8 to 63) to 138 (95% CrI 111 to 167) deaths per 1000 live births. U5M was highest in neighbourhoods of the central urban core and industrial areas, with an average of 95 deaths per 1000 live births across these neighbourhoods. Peri-urban neighbourhoods performed better, on average, but rates varied more across neighbourhoods compared with neighbourhoods in the central urban areas. U5M was negatively correlated with multiple indicators of improved living and socioeconomic conditions among peri-urban neighbourhoods. Among urban neighbourhoods, correlations with these factors were weaker or, in some cases, reversed, including with median household consumption and women's schooling.

**Conclusion** Reducing child mortality in high-burden urban neighbourhoods in GAMA, where a substantial portion of the urban population resides, should be prioritised as part of continued efforts to meet the Sustainable Development Goal national target of less than 25 deaths per 1000 live births.

## Strengths and limitations of this study

► We accessed the full microdata of the latest census (2010) in Ghana that contained birth history data for over 700 000 women aged 25–49 years living in the Greater Accra Metropolitan Area.

► Georeferenced census data allowed for estimation of under-five mortality at the fine spatial scale of neighbourhood, with full coverage of the Greater Accra Metropolitan Area.

► We used a flexible Bayesian spatiotemporal model that allowed each neighbourhood's mortality estimate to be informed by its own data and that of surrounding neighbourhoods, and incorporated weights for the number of births recorded.

► Under-five mortality was indirectly calculated from the summary birth history data because complete birth and death registration data were unavailable. These data may be subjected to recall errors.

► No data were available on the cause of death.

## BACKGROUND

Recent decades have delivered marked reductions in child mortality across all world regions.[1] Despite improvements, almost 8% of all deaths globally in 2019 were children under five years of age,[2] mostly due to preventable and treatable causes linked to infection and malnutrition.[3] The rate of under-five mortality (U5M) in sub-Saharan Africa (SSA) far exceeds other regions, estimated at 76 deaths per 1000 live births in 2019, and under-five deaths in the region account for a growing proportion of the global total.[1]

On average, children living in cities across low-income and middle-income regions have a survival advantage over their rural counterparts, largely due to improved education, employment and healthcare opportunities in urban areas.[4–6] While still predominantly

rural, SSA has the world's fastest-growing urban population and cities are expected to absorb over 75% of the region's population growth over the next three decades.[7] This offers many opportunities for continued improvement in child mortality, however, infrastructure and basic service provision remain major challenges for cities in the region, exacerbated by the rapid pace of urban population growth. Over half of the urban population—more than in any other world region—lives in slums and informal settlements that are often characterised by poverty and concentrated deprivation.[8]

The social determinants of health refer to the conditions in which people are born, grow, live, work and age and have important influence on health inequalities.[9 10] Social gradients in the health of children are well documented, whereby children born into deprivation have lower chances of survival and prosperity.[9–11] In cities, health outcomes and their social, economic and environmental determinants can vary dramatically between households and neighbourhoods.[12 13] The mortality gap between children living in slum versus non-slum urban areas in SSA, for example, can be as large as the gap between rural and urban children.[6 14–16] Inadequate housing, electricity and clean fuel access, water and sanitation facilities, nutrition and healthcare services are among the pathways through which low income or education levels, among other factors termed 'social stratifiers',[9] can increase the susceptibility or hazardous exposures of those most deprived. These mechanisms can act at the individual and household level or at the area level, whereby people with the fewest means are spatially sorted into neighbourhoods with the poorest infrastructure, known as segregation.[17 18] This in turn can contribute to intraurban health inequalities, including in child mortality, seen at the small area level, where advantage tends to cluster.[13 16 19 20]

There is substantial evidence that health outcomes vary at small spatial scales[21–26] and local neighbourhood factors are increasingly recognised as important drivers of population health inequalities.[21 27–30] Subnational estimates of child mortality are often at too course of a scale to capture local variation across cities.[22 31 32] Increased knowledge of the spatial heterogeneity of child mortality within cities at finer scales in SSA countries is necessary to target interventions and programmes towards high-risk populations, accelerate progresstoward the United Nations Sustainable Development Goal (UN SDG) target of less than 25 deaths per 1000 live births for all countries by 2030 (Goal 3, Target 3.2)[33] and better understand the determinants of such inequalities.[34]

With unique access to the complete records of the most recent Ghana Population and Housing Census,[35] we aimed to estimate rates of U5M at the neighbourhood level across the Greater Accra Metropolitan Area (GAMA), providing insight into the magnitude of intraurban inequalities in child mortality within a rapidly growing, low-middle-income city. We quantified U5M rates for 2010, the year of the census, and, aligned with previous studies using census data, examined their relationships

with neighbourhood-level indicators of socioeconomic and living conditions.[21 26 28] This study was conducted within the Pathways to Equitable Healthy Cities study (http://equitablehealthycities.org/).

## METHODS
### Study setting

Ghana is among the most urbanised countries in SSA with an estimated urban population of over 18 million in 2021 (58% of the total population), that is growing by ~3% each year.[7] GAMA is Ghana's administrative and economic capital and accounted for 29% of the country's urban population in 2010.[35] It covers ~1500 km² on the southern coast of the Greater Accra region. According to the 2010 census, GAMA comprises 5019 enumeration areas (EAs)—the smallest administrative geographical unit in Ghana—nested within 406 localities and 12 districts or 'municipalities' (figure 1).

The centrally located Accra Metropolitan Area (AMA)—together with the more heavily industrialised Tema and Ashaiman municipalities to the southeast—contain the most densely populated neighbourhoods. AMA contains the central business district and functions as the city's commercial, industrial and administrative centre.[36 37] Rapid development since Ghana's independence in 1957 has contributed to increased congestion in AMA's residential areas. Planned residential neighbourhoods in AMA remain as legacies of the colonial era, while migrants and low-income individuals have been pushed into slums and other low-income neighbourhoods that can lack basic services and infrastructure.[17] Tema is GAMA's planned industrial hub with structured housing developments and services, and was the fastest-growing municipality following independence. A small fraction (~5%) of GAMA's population lives in areas classified in the 2010 census as rural, mostly in northern GAMA and predominantly in the Ga West and Ga South districts. These districts are characterised by sprawling urban development with high rates of population growth since the 1970s due to congestion of the city centre.[36 38]

Overall, the U5M rate in Ghana almost halved from 1990 to 2010, though considerable subnational inequality persisted.[1 31 32] During this period, the government implemented several national health policies and programmes to improve the use and delivery of maternal and child healthcare services.[39] The National Health Insurance Scheme (NHIS) provides free healthcare to participants[40] and, in 2008, enrolment was made free of charge for pregnant women and children under 18 years of age.[41] Antenatal and postnatal visits, facility delivery (including emergency obstetric care) and neonatal care are all included under the scheme. The User Fees Exemption for Delivery Care was scaled up in 2005 and exempts pregnant women who are not enrolled in NHIS from paying delivery fees.[42] Ghana's 2007–2015 Child Health Policy aimed to unify fragmented programme delivery under a recommended continuum of care for mothers

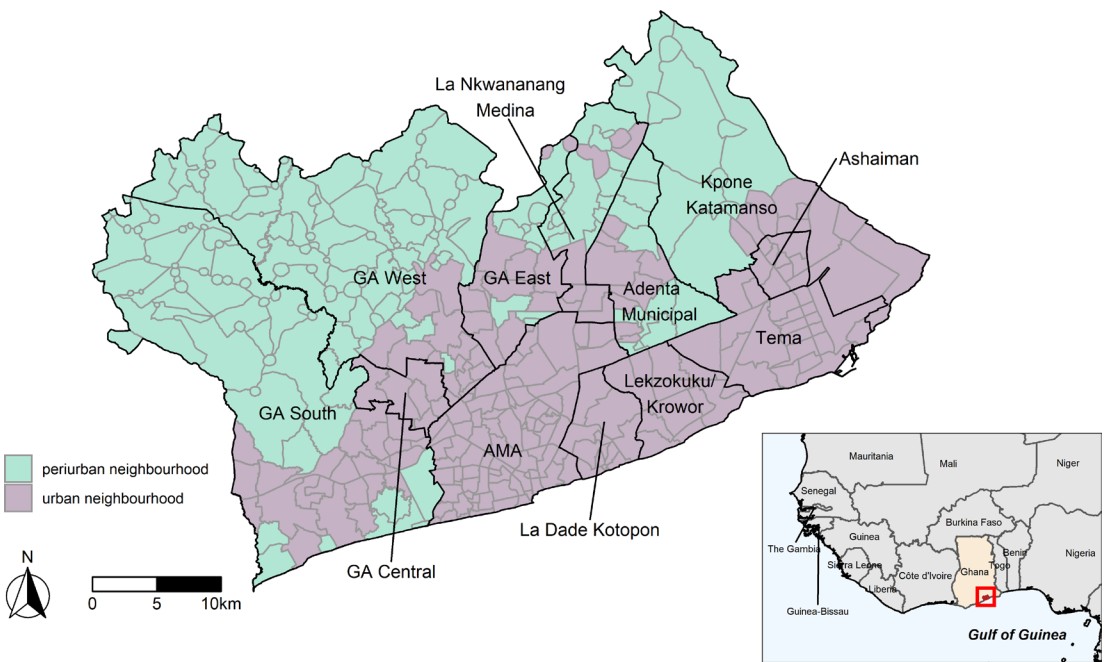

**Figure 1** The Greater Accra Metropolitan Area (GAMA) with neighbourhood boundaries shown in grey and district boundaries shown in black (grey) (source: Ghana Statistical Service). Urban neighbourhoods are shown in purple and peri-urban neighbourhoods shown in green. The inset shows the location of GAMA in Ghana and Western Africa.

and children, scaling up interventions with proven efficacy to prevent child deaths, including, for example, oral rehydration therapy and zinc for treatment of diarrhoea, vitamin A supplementation and antibiotic treatment for pneumonia.[43] Together, these efforts have contributed to reductions in overall child mortality and in inequalities between subregions, though wider concerns have persisted over the quality of care.[44] In Accra, women enrolled in NHIS were found more likely to seek formal care and visit clinics, but enrolment rates were lower (under 35%) among women of childbearing age compared with women over the age of 50.[41] Although most births (>90%) in Greater Accra take place in a health facility in the presence of a skilled health professional, the coverage rate of age-appropriate vaccines in children drops with age (from 76% to 48% comparing children aged 1 vs children aged 2–3 years, respectively). This indicates considerable variability in continued access to care for children.[45]

## Data

We accessed the full microdata of the most recent Ghana Population and Housing Census, conducted in 2010, via the Ghana Statistical Service. The census collected information on the number of children born and surviving at the time of survey, known as their summary birth history, for all women aged 12 years and older. Other individual and household characteristics captured in the census include employment status, occupational industry, schooling level, literacy, household amenities (including type of cooking fuel, drinking and non-drinking water source, sanitation facilities, lighting source and waste disposal method in use), dwelling type and structural features (including roof, floor and wall materials) and EA of residence. Together, these provide information on the socioeconomic and living environments.

We obtained a shapefile from the Ghana Statistical Service with all GAMA EAs, localities and districts geocoded according to the 2010 census geographies. Localities within GAMA were the neighbourhood units used in our analysis, each containing between 1 and 95 EAs. We linked the census data and the shapefile using codes that uniquely identified EAs to determine each individual's neighbourhood of residence. Neighbourhoods were defined by the Ghana Statistical Service and are the administrative units at which urban versus rural classification is defined in Ghana; those with 5000 inhabitant or more are considered urban, and rural otherwise. Some neighbourhoods were recently subdivided due to population growth and, thus, did not meet the urban population threshold despite the urban designation of their constituent EAs in the census. We, therefore, classified GAMA neighbourhoods as urban or periurban according to the historic census-derived urban–rural designation of their constituent EAs. Most neighbourhoods (98%) comprised exclusively urban or exclusively rural EAs. We classified neighbourhoods that contained both urban and rural EAs as urban if over 50% of the population lived in urban EAs and periurban otherwise (figure 1). Ghana does not have an official definition that distinguishes periurban from rural neighbourhoods; however, we used the term periurban to better describe 'rural' neighbourhoods that are located within the administrative border of GAMA on the periphery of the densely populated inner-city and industrial areas (figure 1).

To assess U5M, we summarised the birth history data of women of reproductive age (15–49 years) by five-year age group and neighbourhood. The Maternal Age Cohort (MAC) method was used to estimate the neighbourhood probability of death for children before the age of five (5q0) for each five-year age group, based on the number of children ever born, proportion of children who have died and average parity.[46] The MAC method outperforms alternative methods for estimating U5M from summary birth history data for subnational populations.[47] Each 5q0 was assigned to a reference year prior to the census, using maternal age as a proxy measure for duration of exposure to risk of death for a child. The assigned 5q0 reference years covered the period from 1990 to 2005. We excluded 5q0 estimates derived from women aged 15–19 and 20–24 years, owing to the low numbers of births recorded for these age groups in many neighbourhoods, which could lead to spurious fluctuations in the 5q0 estimates, especially in the more sparsely populated periurban areas. Notably, this is common practice when using demographic methods to estimate population U5M rates.[46–48] This left five 5q0 estimates for each neighbourhood (one derived from each five-year age group of women aged 25–49 years).

## Statistical analysis

To obtain neighbourhood estimates of 5q0 for 2010, the year of the census, we fitted a Bayesian spatio-temporal model to the MAC-derived 5q0 estimates across all 2030 neighbourhood-reference year units, transformed to the probit scale. The model included a linear time trend that could vary by neighbourhood. The time trend allowed data from different reference years, each of which is associated with a different age group, to inform the 5q0 in 2010. The neighbourhood intercepts and slopes were modelled using the Besag, York and Mollié model,[49] where information is shared locally (ie, among adjacent neighbourhoods) through spatially structured random effects with a conditional autoregressive prior and globally through spatially unstructured Gaussian random effects. Neighbourhood-specific intercept and slope values were estimated by the sum of their respective spatially structured and spatially unstructured random effects. The prior distributions in the Bayesian framework allow the neighbourhood-specific parameters to be estimated by a neighbourhood's own data and data of contiguous neighbourhoods. This approach balances overly unstable within-neighbourhood estimates and overly simplified aggregate estimates for all of GAMA. The reported estimate for the wealthy Ringway neighbourhood in AMA is informed entirely by data in bordering neighbourhoods as the data on child deaths were considered implausible (see online supplemental appendix 1). To account for excess variability resulting from small numbers of children born to women in a given age group and neighbourhood, we included a weighted variance term that gave more weight to estimates derived from a higher number of births. Samples from the posterior distributions of the intercepts and slopes were used to estimate 5q0 for the year 2010.

To avoid infinite values on the probit scale, we adjusted all MAC-derived 5q0 estimates of zero (n=146; 7%) to half the minimum estimated non-zero value across all units (0.00316). We conducted sensitivity analyses to ensure that our results were robust to this choice by replacing zero estimates with a lower value of 0.0001 and with the minimum estimated non-zero 5q0.

We monitored convergence using trace plots and obtained 5000 post burn-in samples from the posterior distributions of model parameters. We summarised the distributions of neighbourhood-specific parameters to report neighbourhood U5M estimates and mean U5M across neighbourhoods within districts for 2010, with 95% credible intervals (CrI) that represent the mean and the 2.5th and 97.5th percentiles of the posterior samples, respectively. We present neighbourhood U5M estimates as deaths per 1000 live births.

We calculated neighbourhood-level summary statistics of individual and household characteristics to provide context for our mortality results. We used the within-neighbourhood median household consumption as a measure of neighbourhood socioeconomic level. Household consumption is considered a better indicator of living standards than household income in low-income and middle-income settings.[50 51] The census did not include consumption data, so we used small-area estimation methods to indirectly calculate consumption based on household characteristics described in detail elsewhere.[52] Briefly, we used the 2012 Ghana Living Standards Survey to develop a statistical relationship between household characteristics and consumption. Then using those same household characteristics, we predicted consumption for households in the census. We additionally calculated population density; the proportion of the women of reproductive age (15–49 years) who were literate, had schooling to at least primary, middle, secondary and postsecondary levels; the proportion of the working age (15–64 years) population in any employment and in primary, secondary and tertiary sector occupations; and the proportion of households with indicators of improved living conditions (including dwelling type, materials of flooring, roofing and walls, methods of solid and liquid disposal, type of toilet facility, type of cooking fuel and type of drinking and other water source). Details of our classification of the census responses into indicators of 'improved' versus 'unimproved' living conditions are provided in online supplemental appendix 2.

We measured the correlations between neighbourhood U5M and neighbourhood socioeconomic and living environment indicators using the non-parametric Spearman's rank method. We measured the correlations across all GAMA neighbourhoods and separately across urban and periurban neighbourhoods.

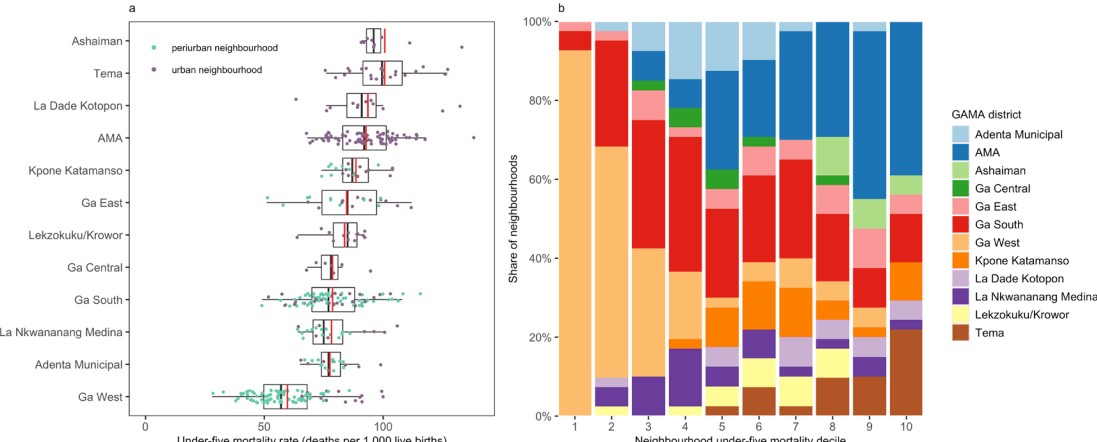

**Figure 2** (A) Neighbourhood under-five mortality in the Greater Accra Metropolitan Area (GAMA) in 2010, by district. The black vertical line and box show the median and IQR of under-five mortality across neighbourhoods in a district and the red vertical line shows the mean. (B) The contribution of neighbourhoods in each district to each decile of neighbourhood under-five mortality. Decile 1 groups the 10% of neighbourhoods with the lowest under-five mortality rates and decile 10 groups the 10% of neighbourhoods with the highest under-five mortality rates.

All analyses were implemented in the open-source statistical software R V.3.6.1. The Bayesian model was implemented using the NIMBLE package V.0.9.1.

### Patient and public involvement

The study used secondary data only.

### RESULTS

There were 713 581 women aged 25–49 years living in GAMA in 2010, who reported a total of 1 425 008 children, 1 312 030 (92.1%) of whom were alive at the time of the census. The number of women in this age range per neighbourhood ranged from 15 to 16 061 (median=564). The proportion of their children born who had died ranged from 0% to 20% across neighbourhoods.

In 2010, the mean neighbourhood U5M rate in GAMA was 80 deaths per 1000 live births (95% CrI 76 to 84). This compares to an estimated 69 deaths per 1000 live births nationally in Ghana.[1] Across all of GAMA, neighbourhood U5M varied almost five-fold, ranging from 28 deaths per 1000 live births (95% CrI 8 to 63) in the Fantsenkor neighbourhood located in the north of GAMA to 138 deaths per 1000 live births (95% CrI 111 to 167) in the Mamobi neighbourhood in the city's urban core (figures 2A and 3). The variation was higher across periurban neighbourhoods compared with across urban neighbourhoods, ranging from 28 to 116 versus from 52 to 138 deaths per 1000 live births, respectively.

We found substantial variation in U5M between and within GAMA's 12 districts. The within-district mean neighbourhood U5M was highest at 101 deaths per 1000 live births in the Tema (95% CrI 89 to 113) and Ashaiman (95% CrI 89 to 114) municipalities situated in the southeast and east of GAMA, respectively. Both municipalities comprised only urban neighbourhoods. The within-district mean neighbourhood U5M was lowest in the

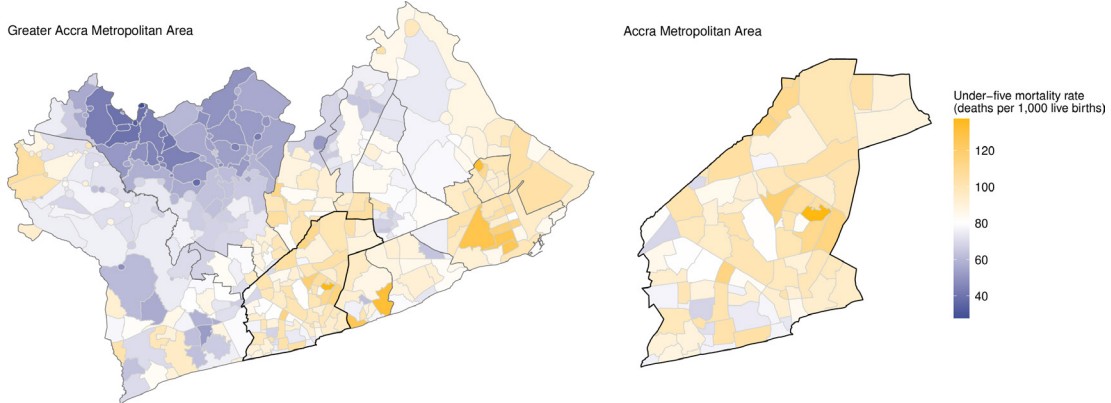

**Figure 3** Under-five mortality rates in neighbourhoods of the Greater Accra Metropolitan Area (GAMA) in 2010. Neighbourhood boundaries are shown in grey, and district boundaries are shown in black. The Accra Metropolitan Area (AMA) bournday is emphasised. The colour scale diverges at the mean under-five mortality rate across all GAMA neighbourhoods. Neighbourhoods with U5M above the GAMA average are shown in orange; neighbourhoods with U5M below the GAMA average are shown in purple.

northern Ga West district (60 deaths per 1000 live births, 95% CrI 52 to 68), where over 70% of neighbourhoods were considered periurban (figure 2A). Although Ga West had the lowest mean mortality among GAMA districts, it had the highest within-district inequality (measured as the relative difference between the neighbourhood with the highest vs lowest mortality), with a 3.6-fold difference between the worst- and best-performing neighbourhoods. Within-district inequality was lowest in the Ga Central and Kpone Katamanso municipalities (1.4-fold difference across neighbourhoods).

The 10% of neighbourhoods with the lowest U5M were concentrated in just three neighbouring districts in western GAMA, namely, Ga West, Ga East and Ga South. The 10% of neighbourhoods with the highest U5M were more dispersed, present in eight of the 12 GAMA districts, though over a third of the highest mortality neighbourhoods fell within the urban core of AMA (figure 2B). Neighbourhood mortality was consistently high in AMA, where 80% of neighbourhoods (n=64) had U5M rates higher than the mean across GAMA neighbourhoods. This was true also for the Tema and Ashaiman municipalities, where 95% and 100% of neighbourhoods, respectively, had higher than average U5M (figure 3). The spatial pattern of U5M reported was unchanged in our sensitivity analyses (online supplemental appendix 3). The mean absolute difference in neighbourhood U5M estimates was 2.24 deaths per 1000 live births in our analysis replacing zero estimates of 5q0 with 0.0001 and −0.73 in our analysis replacing zero estimates with the minimum estimated non-zero 5q0 value.

The correlations between neighbourhood U5M and indicators of improved living and socioeconomic conditions in periurban areas were distinct from urban neighbourhoods located mostly in the urban core and industrial areas (table 1). In periurban areas, U5M was inversely correlated with most of the indicators of improved neighbourhood conditions analysed. The strongest associations were with the proportions of neighbourhood residents living in houses built with improved wall and roof materials, and with an improved lighting source, followed by the share of women aged 15–49 with at least primary or middle school education or who were literate (Spearman's ρ: −0.41 to −0.50). By comparison, correlations were weaker or reversed in urban neighbourhoods. Similar inverse correlations, though smaller in magnitude (Spearman's ρ: −0.14 to −0.23), were found with the share of working age population engaged in secondary or tertiary sector occupations and the proportion of residents living in housing with improved wall and roof materials. No relationship was found for several indicators, including levels of women's literacy and access to improved drinking water source (Spearman's ρ: 0.12 and −0.02, respectively). For others, there was a positive correlation, most notably with indicators of socioeconomic status, for example, the share of women educated to middle school level or above and average household consumption (Spearman's ρ: 0.14 to 0.35).

## DISCUSSION

Our high spatial resolution analysis of child mortality in GAMA, one of the largest metropolitan areas in SSA, revealed considerable intraurban inequality in U5M, with the highest levels in neighbourhoods located in the city's urban core and more heavily industrial areas. Notably, even GAMA's lowest mortality neighbourhood had a mortality rate equivalent to the Dominican Republic and Bhutan, which ranked 129th and 130th, respectively, among countries and territories worldwide in the UN's most recent estimates.[1] At present, no country has an estimated average child mortality rate as high as GAMA's highest mortality neighbourhood.

A general pattern of higher child mortality extended across inner-city and more industrial neighbourhoods, despite higher median household consumption and generally higher levels of post-primary education among women in these neighbourhoods relative to their periurban counterparts. Moreover, child mortality tended to be higher in better-off neighbourhoods of urban GAMA. The spatial concentration of people across a wide socioeconomic spectrum may generate what has been termed 'negative health externalities',[13] whereby localised inequalities compromise the real or perceived capacity of low-resourced individuals to sustain good health. Households that are better off may be negatively impacted by environmental degradation that is not bounded by household or neighbourhood boundaries, for example, poor community sanitation or air pollution from neighbours' solid waste burning or household use of solid fuel stoves.[53–55] It is important to highlight, however, that we cannot draw firm conclusions regarding the harmful impact of localised socioeconomic inequalities on the health of individuals within neighbourhoods from these observed correlations in our study. Associations at the area level that include neighbourhoods with diverse populations may simply obscure any association between wealth and health operating at the individual or household level.

Child mortality can be seen in some places to follow historical patterns of socioeconomic segregation in central Accra.[17] The wealthier Cantonments neighbourhood—home to European settlers in the colonial era—was among the pockets of lower-than-average U5M (63 deaths per 1000 live births), contrasted with higher rates found in low-income or 'slum' neighbourhoods, including Mamobi, Nima and New Town (83–138 deaths per 1000 live births). However, other low-income neighbourhoods had U5M rates below the GAMA average, for example, Jamestown and Sabon Zongo. The health penalty of areas with concentrated poor living conditions that arise from rapid, unplanned population growth, compiled by underinvestment in public services and infrastructure, in SSA cities is well established and supported by previous studies conducted in central Accra and Nairobi.[6 14 56–59] The high mortality rates in city slums in Kenya is thought to have contributed to stalled progress in urban compared with rural areas.[59] The variation seen across poor areas suggests distinct characteristics of these neighbourhoods

**Table 1** Population and household characteristics of neighbourhoods in the Greater Accra Metropolitan Area and correlations with neighbourhood under-five mortality in 2010

| Neighbourhood population and household characteristics | Median (inter-quartile range) | | | Correlation coefficient (Spearman's rho) | | |
|---|---|---|---|---|---|---|
| | All | Periurban | Urban | All | Periurban | Urban |
| Child mortality rate (deaths per 1000 live births) | 80 (68–92) | 69 (57–79) | 90 (79–99) | n/a | n/a | n/a |
| Population | 2972 (767–12798) | 730 (439–1380) | 11754 (6254–20873) | n/a | n/a | n/a |
| Population aged under 5years | 384 (107–1430) | 105 (64–195) | 1267 (703–2486) | n/a | n/a | n/a |
| Median household consumption (GHS) | 5599 (4249–6894) | 4485 (3616–5432) | 6690 (5633–7473) | 0.44* | −0.18* | 0.35* |
| Population density (per km²) | 1858 (614–6366) | 682 (156–1405) | 5610 (1914–14513) | 0.44* | 0.04 | 0.09 |
| Women's literacy† | 90.2% (83.2%–93.3%) | 88.0% (68.9%–93.0%) | 91.4% (88.2%–93.4%) | 0.03 | −0.41* | 0.12 |
| Women's schooling: at least primary† | 90.2% (83.2%–93.3%) | 88.0% (68.9%–93.0%) | 91.4% (88.2%–93.4%) | 0.03 | −0.41* | 0.12 |
| Women's schooling: at least middle† | 78.7% (67.9%–83.4%) | 74.9% (43.3%–81.9%) | 80.4% (76.0%–84.2%) | 0.09 | −0.41* | 0.14* |
| Women's schooling: at least secondary† | 36.6% (24.5%–44.3%) | 30.7% (6.4%–39.8%) | 40.6% (33.3%–47.7%) | 0.18* | −0.38* | 0.21* |
| Women's schooling: post-secondary† | 10.5% (4.9%–15.4%) | 7.4% (1.2%–12.5%) | 12.6% (8.3%–17.6%) | 0.18* | −0.30* | 0.17* |
| Employment‡ | 67.4% (64.7%–70.5%) | 68.2% (64.5%–72.9%) | 67.1% (64.9%–68.9%) | −0.01 | 0.31* | −0.18* |
| Occupation: primary sector‡ | 2.0% (1.5%–3.8%) | 3.4% (1.6%–21.4%) | 1.8% (1.4%–2.5%) | −0.04 | 0.44* | −0.03 |
| Occupation: non-primary sector‡ | 64.0% (60.3%–66.6%) | 61.7% (37.5%–65.9%) | 64.9% (62.6%–66.7%) | 0.04 | −0.25* | −0.14* |
| Non-agricultural household§ | 94.4% (88.8%–96.4%) | 89.1% (47.2%–95.3%) | 95.3% (93.3%–96.9%) | 0.10* | −0.40* | 0.03 |
| Improved dwelling type§ | 88.1% (79.4%–93.4%) | 88.5% (78.1%–94.7%) | 88.1% (80.0%–92.3%) | −0.13* | −0.28* | 0.02 |
| Improved drinking water source§ | 95.1% (85.4%–97.9%) | 92.8% (78.6%–99.0%) | 95.9% (89.7%–97.6%) | −0.11* | −0.39* | −0.02 |
| Improved water source§ | 89.5% (71.8%–96.7%) | 80.5% (50.6%–94.6%) | 93.7% (81.7%–97.1%) | 0.12* | −0.18* | 0.07 |
| Improved toilet facilities§ | 70.2% (48.3%–84.6%) | 74.2% (51.2%–89.7%) | 66.5% (47.3%–80.6%) | −0.18* | −0.29* | 0.00 |
| Improved solid waste disposal§ | 77.4% (54.2%–90.8%) | 64.4% (36.6%–86.3%) | 85.7% (68.6%–93.4%) | 0.34* | 0.05 | 0.26* |
| Improved liquid waste disposal§ | 26.3% (10.2%–40.0%) | 12.7% (1.9%–26.5%) | 32.9% (23.8%–48.3%) | 0.31* | −0.25* | 0.27* |
| Improved cooking fuel use§ | 45.0% (27.4%–55.2%) | 35.9% (4.8%–50.2%) | 49.2% (39.4%–57.9%) | 0.12* | −0.37* | 0.13* |
| Improved lighting source§ | 83.9% (71.9%–92.0%) | 79.8% (55.2%–87.6%) | 88.4% (77.6%–93.3%) | 0.05 | −0.48* | 0.10 |
| Improved floor material§ | 90.9% (84.6%–94.1%) | 92.1% (84.4%–96.2%) | 90.3% (84.9%–93.0%) | −0.14* | −0.18* | −0.05 |
| Improved roof material§ | 95.6% (91.2%–97.2%) | 95.2% (79.8%–98.3%) | 95.6% (93.6%–96.7%) | −0.25* | −0.43* | −0.23* |
| Improved wall material§ | 87.2% (76.0%–92.4%) | 85.9% (59.2%–93.8%) | 87.6% (83.0%–91.7%) | −0.22* | −0.50* | −0.18* |

Median levels across all, periurban and urban neighbourhoods presented with IQR in parentheses. Spearman's rank correlation coefficients show the correlation between neighbourhood under-five mortality and neighbourhood characteristics that indicate the socioeconomic and living conditions. Online supplemental appendix 4 shows the distribution of under-five mortality across quintiles of analysed indicators.
*Significant at the 5% level.
†Measured as the proportion of women of childbearing age (15–49 years) in the neighbourhood.
‡Measured as the proportion of population of working age (15–64 years) in the neighbourhood.
§Measured as the proportion of population in the neighbourhood.
GHS, Ghanaian Cedi.

and vulnerabilities of the people living in them.[53 60–63] Lower U5M rates in some of these neighbourhoods may reflect the success of targeted interventions to reduce mortality in the poorest communities.[64]

We observed the lowest rates of U5M in periurban neighbourhoods, where residents can still benefit from the urban services that contribute to improved child survival in cities, but avoid the potential hazards of inner-city living. A healthy selection effect may also contribute to the observed pattern of U5M, where recent development of periurban neighbourhoods has encouraged wealthier residents to move out of the inner city in search of better standards of living. Some of the areas considered periurban in the Ga districts are more recently developed neighbourhoods mostly occupied by middle-income and high-income individuals. Our finding of lower U5M in neighbourhoods with improved socioeconomic and living conditions in the periurban area is consistent with the established social gradient in child mortality[9–11] and with evidence linking the neighbourhood environment to population health inequalities.[21 27–30] It may reflect greater access to services including healthcare for women in these neighbourhoods, which can influence the health of mothers and their children. However, these reported ecological correlations only provide broader context to the observed spatial patterns of U5M across GAMA and should not be interpreted as causal.

A notable strength of our study is the fine spatial resolution of our U5M estimates, enabled by access to the full microdata of the 2010 census, including birth histories of over 700 000 women. Previous knowledge of child mortality inequalities in the metropolitan area was limited to broad district aggregates[31 32] or smaller areas in the urban core,[61 65] thereby excluding almost half of GAMA's total population, including residents of the rapidly developing Ga districts and the industrial Tema area.[66] In the absence of complete birth and death registration, summary birth history data used in our study can produce robust estimates of U5M[67] that are comparable to estimates directly calculated from complete birth history data that require detailed questionnaires.[47 68] The ease of summary birth history collection allows for its inclusion in national censuses and, in turn, analysis of U5M at finer spatial scales than is achievable through sample surveys.[69] Our use of census data also enabled straightforward linkage of mortality outcomes with data on socioeconomic and living conditions at a common, local spatial scale and avoided issues of sample representativeness. Specifically, the data analysed were of all women aged 25–49 years in GAMA at the time of enumeration, and included those living in regular housing (n=696 279), homeless women/outdoor sleepers (n=13 950), women who were in schools, hospitals, army and service barracks and prisons (n=2361) and a small proportion from other locations (n=991). People were not excluded from the census or this analysis based on migration status. Our analysis makes important methodological advancements from available neighbourhood-level U5M estimates in

GAMA.[61 65] First, we used a validated method for the use of summary birth history data in U5M estimation.[46 47] Second, we applied established Bayesian methods widely used in small area estimation[49] that allowed sharing of information across spatial units to make robust estimates for our study neighbourhoods.

Our results show a different spatial pattern of U5M across GAMA compared with previous estimates reported at the district level that indicated higher U5M in the largely periurban Ga West district and minimal variability across the rest of GAMA.[31] This inconsistency could be due to a scarcity of data from areas outside the urban core used in the previous analysis. Their results rely heavily on data from the Demographic and Health Surveys that, for example, in 2008, sampled 41 women living in only four EAs in Ga West compared with 299 women living in 38 EAs in the AMA. Estimates for Ga West would, therefore, be informed by data with limited geographical coverage within the district and data from bordering rural areas, where child mortality is on average higher.[6] By comparison, our study included data on children's births and deaths from 41 270 women living in all 92 neighbourhoods of GA West.

Our study has several limitations to be considered for future studies. Changes in GAMA over the last decade may mean that our results do not reflect current patterns of U5M. Data from more recent sample surveys in GAMA, however, lack the coverage, scope and sample sizes of the census that enabled us to conduct our analysis at the neighbourhood level. The georeferenced data in the 2021 Ghana census, which is currently in progress, can be used to provide up-to-date estimates of U5M within GAMA and examine changes since 2010. As is the case for all studies of child mortality using census or sample survey data, we were limited to data on neighbourhood of residence at the time of the census and could not account for relocation since the death of a child. We also acknowledge that intervention efforts to prevent child mortality would benefit from more detailed information on the underlying causes of these deaths and age at which GAMA children are most vulnerable.[70 71] In our study, we were unable to separately estimate neonatal, infant or cause-specific mortality rates with the available methods for mortality estimation from summary birth history data and are unaware of any other data that are representative at the neighbourhood level and would enable this analysis. Investment in national civil registration systems across SSA, including in Ghana, could provide these data in real time and enable governments to monitor changes and progress in lowering child mortality and reducing inequalities.[72 73] Finally, we highlight that although we report U5M at a fine-spatial scale, populations within neighbourhoods may still be heterogeneous in their child mortality risk and its determinants, particularly within the more populous inner-city neighbourhoods. This is often discussed in the context of the Modifiable Areal Unit Problem, a statistical bias that can arise when area-level measurements are sensitive to the scale or zoning scheme used.[74]

## CONCLUSIONS

Our city-scale results contribute to a small but growing number of studies showing that within-city variation in child mortality in SSA can be as large as the difference between urban and rural areas or between countries.[6 14 16] Global targets for child mortality, including SDG 3.2, and related monitoring efforts, focus entirely on national mortality rates.[33 75] While important for benchmarking countries' overall performance, progress towards SDG 3.2 does not automatically benefit high mortality and vulnerable population subgroups within countries.[31 76] The heterogeneity of U5M that we found across GAMA neighbourhoods motivates more localised and increasingly disaggregated information on child deaths in order to deliver effective interventions and continue progress toward meeting national and international targets in a region with relatively limited public health resources.

The determinants of child mortality are multifaceted, operate at multiple levels and can also interact with one another. As Ghana continues to urbanise, its cities and metropolitan areas, including GAMA, will play an increasingly important role in building on national child survival efforts. The heterogeneity in child mortality across Accra's neighbourhoods highlights the need for an explicit focus on equity in the context of rapid urbanisation in SSA, with an emphasis on the social determinants of health. Concentrated deprivation in households and neighbourhoods can compound the risk of child mortality. Our study identified neighbourhoods with high child mortality in GAMA's central and industrial areas, where the urban poor may still face financial and physical barriers to accessing health services,[41 77] potentially compounding health risks associated with disadvantaged social and living conditions. Universal access to high-quality healthcare services can mitigate mortality inequalities in settings where children are born into different socioeconomic and environmental circumstances.[70 78 79] There are proven, scalable healthcare interventions that reduce U5M, the causes of which are often preventable and/or treatable.[70 80] In many low- and middle-income countries, the rise in facility births has not produced the expected improvements in child mortality, demonstrating the importance of continued access to quality care throughout the early years of life for all children and mothers across the socioeconomic spectrum.[78 81]

Child mortality in periurban neighbourhoods was on average lower than inner-city and industrial areas, but it was also more variable and inversely correlated with characteristics that indicate improved socioeconomic and living environments of neighbourhoods. Complementary investment in developing infrastructure and services in neighbourhoods outside of the urban core, while ensuring that conditions in densely populated central and industrial areas do not deteriorate, could further contribute to improving child mortality and promoting health equity.

**Author affiliations**
¹Department of Epidemiology, Biostatistics and Occupational Health, McGill University, Montreal, Quebec, Canada
²Institute for Health and Social Policy, McGill University, Montreal, Quebec, Canada
³Department of Epidemiology and Biostatistics, School of Public Health, Imperial College London, London, UK
⁴MRC Centre for Environment and Health, Imperial College London, London, UK
⁵Regional Institute for Population Studies, University of Ghana, Accra, Ghana
⁶Department of Environmental Health Sciences, University of Massachusetts Amherst, Amherst, Massachusetts, USA
⁷Ghana Statistical Service, Accra, Ghana
⁸University of Cape Coast, Cape Coast, Ghana
⁹Department of Geography, McGill University, Montreal, Québec, Canada
¹⁰Department of Geography and Resource Development, University of Ghana, Legon, Greater Accra, Ghana
¹¹Institute of Statistical, Social and Economic Research, University of Ghana, Accra, Ghana
¹²Abdul Latif Jameel Institute for Disease and Emergency Analytics, Imperial College London, London, UK

**Acknowledgements** We also thank Tzu-Wei Joy Tseng and Emmalin Buajitti for their help in preparation of the manuscript and helpful discussions during revisions.

**Contributors** HB, ME, AAB, GO and JB developed the study concept. HB, AAB, SKA, JDA, SEM, REA, AC, GO and SAM contributed to data collation. HB conducted the statistical analysis with input from JEB, RA, AMS, BER, ME and JB. HB, AAB, REA, SAM, BER, ME and JB contributed to data interpretation. HB wrote the first draft of the manuscript with input from JB. All authors contributed to revising and finalising the manuscript. HB and JB have primary responsibility for the final content. JB will act as guarantor.

**Funding** This work is supported by the Pathways to Equitable Healthy Cities grant from the Wellcome Trust [209376/Z/17/Z]. HB is supported by a Canadian Institute of Health Research Banting Postdoctoral Fellowship. For the purpose of Open Access, the author has applied a CC BY public copyright licence to any Author Accepted Manuscript version arising from this submission.

**Disclaimer** The funders had no role in the study design, data collection and analysis, decision to publish, or preparation of the manuscript.

**Map disclaimer** The inclusion of any map (including the depiction of any boundaries therein), or of any geographic or locational reference, does not imply the expression of any opinion whatsoever on the part of BMJ concerning the legal status of any country, territory, jurisdiction or area or of its authorities. Any such expression remains solely that of the relevant source and is not endorsed by BMJ. Maps are provided without any warranty of any kind, either express or implied.

**Competing interests** None declared.

**Patient consent for publication** Not applicable.

**Provenance and peer review** Not commissioned; externally peer reviewed.

**Data availability statement** Data are available in a public, open access repository. Data may be obtained from a third party and are not publicly available. Estimates of neighbourhood U5M will be made available at http://equitablehealthycities.org/data-download/. The full microdata of the 2010 Ghana Population and Housing Census are with the Ghana Statistical Service and are not publicly available. Data from a random 10% sample of households enumerated in the 2010 Population and Housing census are publicly available and can be downloaded from Ghana Statistical Services online data catalogue (https://www2.statsghana.gov.gh/nada/index.php/catalog/51).

**ORCID iD**
Honor Bixby http://orcid.org/0000-0002-0513-5292

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
