## [Reviewer comments · BMJ Open]

ARTICLE DETAILS

TITLE (PROVISIONAL)	Quantifying within-city inequalities in child mortality across neighbourhoods in Accra, Ghana: A Bayesian spatial analysis.
AUTHORS	Bixby, Honor; Bennett, James; Bawah, Ayaga; Arku, Raphael; Annim, Samuel; Anum, Jacqueline; Mintah, Samilia; Schmidt, Alexandra; Agyei-Asabere, Charles; Robinson, Brian; Cavanaugh, Alicia; Agyei-Mensah, Samuel; Owusu, George; Ezzati, Majid; Baumgartner, Jill

VERSION 1 – REVIEW

REVIEWER	Ortigoza, Ana Drexel University School of Public Health, Urban Health Collaborative
REVIEW RETURNED	17-Aug-2021

GENERAL COMMENTS	Thank you for inviting me to review this manuscript. This research is very relevant to the study on within city inequalities in health, particularly among children and in the context of different gradient of urban poverty (formal and informal). Although objective of the study is not clearly described, it seems that the main goal of the study is to describe intra-urban inequalities in under-five mortality using within city estimations of mortality. The study made a good work in estimating under five mortality rates for small areas (neighborhood) using an indirect demographic method mostly implemented at national levels. Although the methodological approach for the estimation of mortality looks robust, I have some comments below that might be of author's consideration for clarification of the information and better examination of the data available. I detail my comments by paper sections Introduction: Authors clearly described the challenges related to the lack of information about inequalities in child/under-five mortality in Africa and in Ghana and the need of accounting for within city inequalities in order to advance SDGs. However, since the focus of this study is the Greater Accra Metropolitan Area (GAMA), I consider that the audience would benefit from having more information about the characteristics and dynamics of t GAMA. Since the distribution of wealth in some big metropolitan areas of the Global South may not follow same patterns as in American or European cities it could be helpful to know about these patterns in GANA. Even when it is succinctly mentioned in the discussion that in recent years distribution of wealthy population moved from the core area to the periphery it is not very clear if this is the same context by the period when the analysis is carried out.
---

	I consider objectives should be stated clearer as well as the study period. Since in the result section authors are describing the distribution of mortality by different levels of neighborhood socioeconomic indicators, it could be a good that they established in the introduction the hypothesized distribution of mortality in relation to the neighborhood socioeconomic inequalities. This could then re-captured in the discussion section to analyze whether the results were consistent or not with the a priori hypothesis Methods Authors described that neighborhood were classified as urban or peri-urban according to the census-derived designation of their constituent EAs as urban or rural, respectively. Since the definition of urban and rural could vary by countries, I think it would be helpful to specify the threshold of population size that separate both categories, as well as other characteristics defined by the Ghana Statistical service that may contribute to differentiate them. Further in this section, it is mentioned that 5q0 estimates derived from women aged 15-24 years were excluded from analysis to avoid upward bias of results and reduce instability of estimates due to low numbers of births. I think this could be a source of selection bias that could impact the estimation of under-five mortality as adolescent birth rate may be more frequent in rural (periurban) areas and hence infant and under-five mortality more likely for this maternal age group. I wonder if this is impacting in the lower rates observed for peri urban areas compared to urban areas, in addition to the probability of less accurate registration of census birth history in rural EAs with respect to urban EAs. I may suggest a complementary/ sensitivity analysis incorporating this population age-group to observe the impact this exclusion may have had in the findings described. Additionally, I would suggest incorporating in the limitation section the potential bias that could have been introduced by excluding this population from the analysis. Results/ discussion Since authors are describing mere correlations between neighborhood indicators and mortality I would suggest they would be cautious on describing them as associations between exposures and outcomes, as these correlations may be influenced by other factors not accounted for in this analysis. These caveats should also be considered in the discussion of the results and the comparison with other similar and discordant results. As mentioned in previous comment in the introduction section, it would be interesting that authors could make a clear explanation in the discussion on why results are or not as expected given their hypothesis Finally I may suggest that authors consider also other findings in similar scope of work such as Kimane- Murage 2014 https://doi.org/10.1016/j.healthplace.2014.06.003
--	--

REVIEWER	Tornero Patricio, Sebastián Servicio Andaluz de Salud Area de Gestion Sanitaria de Osuna, Pediatría
REVIEW RETURNED	21-Aug-2021

GENERAL COMMENTS	This investigation is addressing a very important issue related to health: social determinants of health, in a very interesting city and country. So, I do consider it very relevant. I will give my inputs to try to improve the manuscript.
---

	The “background” needs to be developed significantly. It should address de Social Determinant Of Health including some bibliography as Marmot’s investigations and papers based on census as this is based (look for some of the many studies related to this issue using census in USA, UK, Canada or Spain (Medea Research Group). It should include some research about the impact of using different geographic units (zip code vs districts vs neighbourhood vs census units). Also, there should be a deeper description of the city of Accra, their neighbourhood history and the access of health services in each of them. The objective is better described in the abstract that in the manuscript (better using infinitive). Also, it could include “to estimate U5 mortality” as another objective of the study. Methods and Results: It is not completely clear if the differences between U5 mortality are expressed between neighbourhoods or between districts, because the authors are using both sometimes and it is confusing. Despite "house consumption" is used as an indicator of socioeconomic level, the independent variables are geographics (urban vs peri-urban neighbourhoods). I do strongly recommend to use an indicator of socioeconomic level, as house consumption or house condition or education level, to classify the neighbourhoods by socioeconomic factors (deprived or not deprive, for instance). As it is mentioned, using neighbourhoods could lead to mistakes at the analysis, not finding differences when there are census units with high U5 mortality within the neighbourhood or vice versa. The authors should use census units and this sentence should make you seriously consider it: “A general pattern of higher child mortality extended across inner city and more industrial neighbourhoods despite higher median household consumption and generally higher levels of post-primary education among women in these neighbourhoods relative to their peri-urban counterparts. Moreover, child mortality tended to be higher in better-off neighbourhoods of urban GAMA.” This study needs to do an estimation of neonatal mortality because it could explain considerably the differences founded. Also, the access to obstetrics care and maternal health need to be assessed . It also should do an estimation of U5 mortality of the non-censed population (illegal immigrants) and how not including them, could affect the results. Using U5 mean to compare neighbourhoods could cause also the same mistakes described above. The authors could classify neighbourhoods by decils or quintiles, depending on socioeconomic variables, and compare results. Lastly, the conclusion of this investigation give more importance to the health care services than to the social determinants of health, which, in my opinion, it merits a deeper reflection.
--	--

VERSION 1 – AUTHOR RESPONSE

Reviewer: 1

Dr. Ana Ortigoza, Drexel University School of Public Health
Comments to the Author:

Thank you for inviting me to review this manuscript. This research is very relevant to the study on within city inequalities in health, particularly among children and in the context of different gradient of urban poverty (formal and informal).

1. Although objective of the study is not clearly described, it seems that the main goal of the study is to describe intra-urban inequalities in under-five mortality using within city estimations of mortality.

To make sure the objective is more clearly stated. We modified both the abstract (p. 2) and the background (pp. 6-7) that now read as below, respectively:

'We aimed to estimate rates of under-five mortality at the neighbourhood level for Ghana's Greater Accra Metropolitan Area and measure the extent of intra-urban inequalities.'

'With unique access to the complete records of the most recent Ghana Population and Housing Census,¹ we aimed to estimate rates of under-five mortality (U5M) at the neighbourhood level across the Greater Accra Metropolitan Area (GAMA), providing insight into the magnitude of intra-urban inequalities in child mortality within a rapidly growing, low-middle income city. We quantified under-five mortality rates for 2010, the year of the census, and, aligned with previous studies using census data, examined their relationships with neighbourhood-level indicators of socioeconomic and living conditions.²⁻⁴

The study made a good work in estimating under five mortality rates for small areas (neighborhood) using an indirect demographic method mostly implemented at national levels. Although the methodological approach for the estimation of mortality looks robust, I have some comments below that might be of author's consideration for clarification of the information and better examination of the data available. I detail my comments by paper sections:

Introduction

Authors clearly described the challenges related to the lack of information about inequalities in child/under-five mortality in Africa and in Ghana and the need of accounting for within city inequalities in order to advance SDGs.

2. However, since the focus of this study is the Greater Accra Metropolitan Area (GAMA), I consider that the audience would benefit from having more information about the characteristics and dynamics of t GAMA. Since the distribution of wealth in some big metropolitan areas of the Global South may not follow same patterns as in American or European cities it could be helpful to know about these patterns in GANA. Even when it is succinctly mentioned in the discussion that in recent years distribution of wealthy population moved from the core area to the periphery it is not very clear if this is the same context by the period when the analysis is carried out.

As recommended, we included a more detailed description of urban development in GAMA in the study setting section (pp. 7-8):

'The centrally located Accra Metropolitan Area (AMA) together with the more heavily industrialised Tema and Ashaiman municipalities in the east of the city contain the most densely populated

neighbourhoods. AMA contains the central business district and functions as the city's commercial, industrial and administrative centre.^{5 6} Rapid development since Ghana's independence in 1957 has seen AMA's residential areas become increasingly congested. Planned residential neighbourhoods remain as legacies of the colonial era, while migrants and low-income individuals have been pushed into slums and other low-income neighbourhoods lacking basic services and infrastructure.⁷ Tema is GAMA's planned industrial hub with structured housing developments and services, and was the fastest growing municipality following independence. A small fraction (~5%) of GAMA's population lives in areas classified in the 2010 census as rural, mostly in northern GAMA and predominantly in the Ga West and Ga South districts. These districts are characterised by sprawling urban development with high rates of population growth since the 1970s due to congestion of the city centre.^{5 8}

3. I consider objectives should be stated clearer as well as the study period.

Please see response #1 above regarding additional text on the study objectives. We also added information on the study period (p. 6):

'We quantified under-five mortality for 2010, the year of the census.'

In addition, we added text to the data section that describes our application of the Maternal Age Cohort method to derive probabilities of death under five (5q0) from the summary birth history data and assign each to a reference year (p. 10):

'The assigned 5q0 reference years covered the period from 1990 to 2005.'

4. Since in the result section authors are describing the distribution of mortality by different levels of neighborhood socioeconomic indicators, it could be a good that they established in the introduction the hypothesized distribution of mortality in relation to the neighborhood socioeconomic inequalities. This could then re-captured in the discussion section to analyze whether the results were consistent or not with the a priori hypothesis.

As suggested, we included additional information on factors that may contribute to inequalities in child mortality to the Background section of the manuscript that we then revisit in the Discussion. The Background now includes the following (pp. 5-6):

'The social determinants of health refer to the conditions in which people are born, grow, live, work and age and have important influence on health inequities.^{9 10} Social gradients in the health of children are well-documented, whereby children born into deprivation have lower chances of survival and prosperity.⁹⁻¹¹ In cities, health outcomes and their social, economic and environmental determinants can vary dramatically between households and neighbourhoods.^{12 13} The mortality gap between children living in slum versus non-slum urban areas in SSA, for example, can be as large as the gap between rural and urban children.¹⁴⁻¹⁷ Inadequate housing, electricity and clean fuel access, water and sanitation facilities, nutrition and healthcare services are among the pathways through which income or education levels, among other factors termed 'social stratifiers',⁹ can increase the susceptibility or hazardous exposures of those most deprived. These mechanisms can act at the individual and household level or area level, whereby people with the fewest means are spatially sorted into neighbourhoods with the poorest infrastructure, known as segregation.^{7 18} This in turn can contribute to intra-urban health inequalities, including in child mortality, seen at the small area level, where advantage tends to cluster.^{13 17 19 20}

We also added text throughout the discussion to recapture the background information and acknowledge where our results may or may not be inline with any a priori hypothesis for the

relationship between improved socio-economic conditions in neighbourhoods and child mortality. For example, we have included the following (p. 18):

'Our finding of lower U5M in neighbourhoods with improved socio-economic and living conditions in the peri-urban area is consistent with the established social gradient in child mortality,⁹⁻¹¹ and with evidence linking the neighbourhood environment to population health inequalities.^{2 3 21-23}'

5. Authors described that neighborhood were classified as urban or peri-urban according to the census-derived designation of their constituent EAs as urban or rural, respectively. Since the definition of urban and rural could vary by countries, I think it would be helpful to specify the threshold of population size that separate both categories, as well as other characteristics defined by the Ghana Statistical service that may contribute to differentiate them.

As recommended, we edited this section (pp. 9-10) to provide the Ghana-specific definition of urban and rural. In Ghana, neighbourhoods with a population of over 5,000 are considered urban. It now reads:

'Neighbourhoods were defined by the Ghana Statistical Service and are the administrative units at which urban versus rural classification is defined in Ghana; those with 5,000 inhabitant or more are considered urban, and rural otherwise ... Ghana does not have an official definition that distinguishes between peri-urban and rural neighbourhoods; however, we have used the term peri-urban to better describe 'rural' neighbourhoods that are located within the administrative border of the Greater Accra Metropolitan Area on the periphery of the densely populated inner-city and industrial areas.'

6. Further in this section, it is mentioned that 5q0 estimates derived from women aged 15-24 years were excluded from analysis to avoid upward bias of results and reduce instability of estimates due to low numbers of births. I think this could be a source of selection bias that could impact the estimation of under-five mortality as adolescent birth rate may be more frequent in rural (periurban) areas and hence infant and under-five mortality more likely for this maternal age group. I wonder if this is impacting in the lower rates observed for peri urban areas compared to urban areas, in addition to the probability of less accurate registration of census birth history in rural EAs with respect to urban EAs. I may suggest a complementary/ sensitivity analysis incorporating this population age-group to observe the impact this exclusion may have had in the findings described. Additionally, I would suggest incorporating in the limitation section the potential bias that could have been introduced by excluding this population from the analysis.

To address this comment, we edited the manuscript to better explain and justify our approach for the main analysis. Notably, the exclusion of data from younger women is common practice when using demographic methods to estimate population under-five mortality rates, which we have now expressed more clearly in the manuscript. Although the Maternal Age Cohort (MAC) method used in our analysis is an advance on the standard Brass method and attempts to account for the excess mortality in younger ages, the developers of this demographic method still recommend removing estimates from younger ages because of errors associated with the small sample sizes for younger ages.^{24 25} Specifically, where there are small numbers of children born, small differences in the number of children dead can lead to larger differences in the proportion of children dead and hence in estimated under-five mortality rates. The developers thus recommend ignoring data for 15–19-year-olds, however, we have extended this to include 20-24 years owing to the high spatial resolution of our analysis and the small samples sizes in some of our spatial units.

Only ~5% of the GAMA population lived in the peri-urban (rural) neighbourhoods and the number of women and births in these neighbourhoods, especially in younger age groups, was low. In peri-urban areas, there were on average only 4 births per neighbourhood for women aged 15-19 years and 32

births per neighbourhood for women aged 20-25 years, where the average births per woman were 0.07 and 0.5 in these age groups, respectively. Moreover, in peri-urban areas, the total number of children born was below 25 for women aged 15-19 years in 99% of neighbourhoods and in 53% for women aged 20-24 years. These proportions were smaller in urban neighbourhoods, but still elevated compared with older age groups. We therefore excluded data from these two younger age groups from the Bayesian modelling a priori, owing to the insufficient numbers of children born to reliably estimate the neighbourhood 5q0, particularly in the peri-urban areas and when assessing the variation between neighbourhoods.

We now clarify this in the text (p.10), which now reads:

'We excluded 5q0 estimates derived from women aged 15-19 and 20-24 years owing to the low numbers of births recorded for these age groups in many neighbourhoods, which could lead to spurious fluctuations in the 5q0 estimates, especially in the more sparsely-populated peri-urban areas. Notably, this is common practice when using demographic methods to estimate population under-five mortality rates.²⁴⁻²⁶'

To further address the reviewer's concern about bias, we additionally conducted a sensitivity analysis where our analysis including data from all women aged 15-49. The geographical pattern of under-five mortality remains consistent with the results presented from our analysis of data from women aged 25-49 years (see Fig. 1 below), indicating that excluding this group did not introduce significant bias or influence our conclusion that under-five mortality rates were generally lower among peri-urban neighbourhoods than those in the central and industrial areas of GAMA. In fact, in the sensitivity analysis we estimated generally lower under-five mortality in Ga West (peri-urban) neighbourhoods and higher mortality in Accra Metropolitan Area neighbourhoods (see Fig. 2 below), and hence there was an increase in the overall inequality between peri-urban and urban areas.

We did not include this sensitivity analysis in the manuscript since our overall results are the same and it is generally advised by developers of this method not to include younger women using these methods because it can introduce bias. However, we are happy to do so if the Editor feels this would improve the manuscript.

Fig. 1. Under-five mortality rates in 2010 by GAMA neighbourhood. Results are from a sensitivity analysis including data from women aged 15-49 years.

Fig. 2. Comparison of GAMA neighbourhood under-five mortality rates from the analysis including data from women aged 15-49 years versus the results reported from the analysis including data from women aged 25-29 years.

7. Since authors are describing mere correlations between neighborhood indicators and mortality I would suggest they would be cautious on describing them as associations between exposures and outcomes, as these correlations may be influenced by other factors not accounted for in this analysis. These caveats should also be considered in the discussion of the results and the comparison with other similar and discordant results.

To ensure this is clear to readers, we edited the text throughout to avoid presenting our findings as causal associations between exposures and our outcome. The manuscript now also includes an explicit warning against any causal interpretation of the correlation analyses in the Discussion (p. 18):

'these reported ecological correlations are intended only to give broader context to the observed spatial patterns of U5M across GAMA and should not be interpreted as causal.'

Further, we added text to the discussion of the positive correlation between average household consumption and under-five mortality in urban neighbourhoods. We hypothesised a role of within-neighbourhood inequality and the negative impacts on residents across the socio-economic spectrum. We added the following text to warn of over-interpretation of area-level associations (p. 16):

'It is important to highlight, however, that we cannot draw firm conclusions regarding the harmful impact of localised socio-economic inequalities on the health of individuals within neighbourhoods from these observed correlations in our study. Associations at the area level that include neighbourhoods with diverse populations may simply obscure any association between wealth and health operating at the individual or household level.'

8. As mentioned in previous comment in the introduction section, it would be interesting that authors could make a clear explanation in the discussion on why results are or not as expected given their hypothesis.

We added as recommended. Please see response to Comment #4.

9. Finally I may suggest that authors consider also other findings in similar scope of work such as Kimane- Murage 2014 <https://doi.org/10.1016/j.healthplace.2014.06.003>

As recommended, we added reference and discussion of this study in the discussion section of the manuscript where it supports similar findings of high child mortality in urban slums in Accra (p. 17):

'The health penalty of areas with concentrated poor living conditions that arise from rapid, unplanned population growth, compounded by underinvestment in public services and infrastructure, in SSA cities is well established and supported by previous studies conducted in central Accra and Nairobi.^{14 15 27-30} The high mortality rates in city slums in Kenya has contributed to stalled progress in urban compared to rural areas .³⁰

Reviewer: 2

Dr. Sebastián Tornero Patricio, Servicio Andaluz de Salud Area de Gestion Sanitaria de Osuna

This investigation is addressing a very important issue related to health: social determinants of health, in a very interesting city and country. So, I do consider it very relevant. I will give my inputs to try to improve the manuscript.

10. The "background" needs to be developed significantly. It should address the Social Determinant Of Health including some bibliography as Marmot's investigations and papers based on census as this is based (look for some of the many studies related to this issue using census in USA, UK, Canada or Spain (Medea Research Group).

As recommended, we have significantly expanded the background to address the social determinants of health and the contribution to within-city variability in health outcomes (pp. 5-6):

'The social determinants of health refer to the conditions in which people are born, grow, live, work and age and have important influence on health inequities.^{9 10} Social gradients in the health of children are well-documented, whereby children born into deprivation have lower chances of survival and prosperity.⁹⁻¹¹ In cities, health outcomes and their social, economic and environmental determinants can vary dramatically between households and neighbourhoods.^{12 13} The mortality gap between children living in slum versus non-slum urban areas in SSA, for example, can be as large as the gap between rural and urban children.¹⁴⁻¹⁷ Inadequate housing, electricity and clean fuel access, water and sanitation facilities, nutrition and healthcare services are among the pathways through which income or education levels, among other factors termed 'social stratifiers',⁹ can increase the susceptibility or hazardous exposures of those most deprived. These mechanisms can act at the individual and household level or area level, whereby people with the fewest means are spatially sorted into neighbourhoods with the poorest infrastructure, known as segregation.^{7 18} This in turn can contribute to intra-urban health inequalities, including in child mortality, seen at the small area level, where advantage tends to cluster.^{13 17 19 20}

We have also included additional discussion of small area studies to the Background, along with references that use census data, as suggested (p. 6):

'There is substantial evidence that health outcomes vary at small spatial scales^{3 4 31-34} and local neighbourhood factors are increasingly recognised as important drivers of population health inequalities.^{2 3 21-23}

11. It should include some research about the impact of using different geographic units (zip code vs districts vs neighbourhood vs census units).

It is the nature of any spatial analysis that potentially diverse populations are aggregated to report outcomes. The opportunity of small-area estimation methods is that outcomes can be robustly estimated for fine-scale spatial units.¹⁹ Here, we leveraged these techniques to quantify under-five mortality at a higher spatial resolution than was previously available in GAMA and examine the intra-urban inequalities missed by reports of district aggregates.

Nevertheless, we acknowledge that outcomes at the small area level, such as neighbourhood units, can still mask variation within these populations. This is often discussed in the context of the Modifiable Areal Unit Problem (MAUP), a statistical bias that can arise when area-level measurements are sensitive to the scale or zoning scheme used. We have edited the discussion to reflect on this point (p. 20):

'We highlight that although we report under-five mortality at a fine-spatial scale, populations within neighbourhoods may still be heterogenous with regards to their child mortality risk, particularly within the more populous inner-city neighbourhoods. This is often discussed in the context of the Modifiable Areal Unit Problem, a statistical bias that can arise when area-level measurements are sensitive to the scale or zoning scheme used.³⁵

12. Also, there should be a deeper description of the city of Accra, their neighbourhood history and the access of health services in each of them.

As recommended, we expanded the study setting section to include a more detailed description of our study area (GAMA) and its development (pp. 7-8). We have included the modified text in Response #2.

In addition, we added discussion of access to healthcare in Accra. Currently no detailed assessment of health service access exists for neighbourhoods in Accra. We focussed on policies in Ghana that influenced access to maternal and child health services and are thought to have contributed to the

majority (over 90%) of births in Greater Accra taking place in a health facility in the presence of a skilled health professional. The decline in child vaccination rates with age indicates there is variation however not all children have the same access to continued care. The following is now included in the study setting section of the Methods (p.8):

'Overall the under-five mortality rate in Ghana almost halved from 1990 to 2010 although considerable subnational inequality persisted.³⁶⁻³⁸ In this period, the government implemented several national health policies and programs to improve the use and delivery of maternal and child health care services.³⁹ The National Health Insurance Scheme provides free health care to participants⁴⁰ and enrolment has been free of charge for pregnant women and children (under-18 years) since 2008.⁴¹ Ante- and postnatal visits, facility delivery (including emergency obstetric care) and neonatal care are all included under the scheme. The User Fees Exemption for Delivery Care was scaled up in 2005 and prevents pregnant women who are not enrolled in NHIS from paying delivery fees.⁴² Ghana's 2007-2015 Child Health Policy aimed to unify fragmented program delivery under a recommended continuum of care for mothers and children. scaling up interventions with proven efficacy to prevent child deaths, including, for example, oral rehydration therapy and zinc for treatment of diarrhoea, vitamin A supplementation and antibiotic treatment for pneumonia.⁴³ Together these efforts are believed to have contributed to reductions in overall child mortality and in inequalities between subregions though wider concerns have persisted over the quality of care.⁴⁴ In Accra, women enrolled in NHIS were more likely to seek formal care and visit clinics but enrolment rates were lower (under 35%) among women of child-bearing age compared to women over the age of 50.⁴¹ Although, most births in Greater Accra (over 90%) take place in a health facility in the presence of a skilled health professional, the coverage rate of vaccines in children drops with age (from 76% to 48% comparing children aged 1 versus children aged 2-3 years, respectively). This indicates considerable variability in access to continued care.⁴⁵

13. The objective is better described in the abstract than in the manuscript (better using infinitive). Also, it could include "to estimate U5 mortality" as another objective of the study.

As recommended, we have modified the manuscript to state the objective of the study more clearly (pp. 6-7):

'we aimed to estimate rates of under-five mortality (U5M) at the neighbourhood level across the Greater Accra Metropolitan Area (GAMA), providing insight into the magnitude of intra-urban inequalities in child mortality within a rapidly growing, low-middle income city. We quantified under-five mortality rates for 2010, the year of the census, and, aligned with previous studies using census data, examined their relationships with neighbourhood-level indicators of socioeconomic and living conditions.²⁻⁴

14. Methods and Results: It is not completely clear if the differences between U5 mortality are expressed between neighbourhoods or between districts, because the authors are using both sometimes and it is confusing.

As recommended, we have edited the text for clarity. We added text to the methods to state that we report both neighbourhood under-five mortality estimates and the within-district mean under-five mortality calculated across neighbourhoods within each district, together with their 95% credible intervals (p. 12):

'We summarised the distributions of neighbourhood-specific parameters to report neighbourhood U5M estimates for 2010 and mean U5M across neighbourhoods within districts, with 95% credible intervals (CrI) that represent the mean and the 2.5th and 97.5th percentiles of posterior samples, respectively.'

We also modified the results to express more clearly when we report the within-district mean neighbourhood under-five mortality (p. 14):

'We found substantial variation in U5M between and within GAMA's twelve districts. The within-district mean neighbourhood U5M was highest at 101 deaths per 1,000 live births in the Tema (95% CrI 89-113) and Ashaiman (95% CrI 89-114) municipalities situated in the southeast and east of GAMA, respectively. Both municipalities comprised only urban neighbourhoods. The within-district mean neighbourhood U5M was lowest in the northern Ga West district (60 deaths per 1,000 live births, 95% CrI 52-68), where over 70% of neighbourhoods were considered peri-urban (Figure 2a).'

15. Despite "house consumption" is used as an indicator of socioeconomic level, the independent variables are geographics (urban vs peri-urban neighbourhoods). I do strongly recommend to use an indicator of socioeconomic level, as house consumption or house condition or education level, to classify the neighbourhoods by socioeconomic factors (deprived or not deprive, for instance).

We edited the methods to better explain that all indicators of improved neighbourhood living and socio-economic conditions were calculated at the neighbourhood level (p. 12):

'We calculated neighbourhood-level summary statistics of the individual and household characteristics within each neighbourhood to provide context for our child mortality results.'

To further address this comment, we edited the methods to better describe our estimation of median household consumption level for each neighbourhood. Unlike other indicators, household consumption was not directly measured in the census and had to be estimated from the available household characteristics data. The text now reads (p. 12):

'We used the within-neighbourhood median household consumption as a measure of neighbourhood socioeconomic level. Household consumption is considered a better indicator of living standards than household income in low- and middle-income settings^{46 47} The census did not include consumption data, so we used small-area estimation methods to indirectly calculate consumption based on household characteristics described in detail elsewhere.⁴⁸ Briefly, we used the 2012 Ghana Living Standards Survey to develop a statistical relationship between household characteristics and consumption. Then using those same household characteristics, we predicted consumption for households in the census.'

We additionally edited our description of the correlation analysis in the methods to more clearly state that we stratified on urban versus peri-urban neighbourhood status rather than including them as independent variables in our models (p. 13 and below). Stratifying by urban/peri-urban status of neighbourhoods revealed differences in the correlations between under-five mortality and neighbourhood characteristics where indicators of improved neighbourhood conditions were linked to lower mortality only in peri-urban areas. Average household consumption was one such indicator. In other words, deprivation and the experience of it in relation to neighbourhood of residence may have different impacts on child mortality in neighbourhoods of the urban core versus in the periphery. We agree with Comment #16 that this interpretation is only speculative and an analysis at the neighbourhood-level could simply obscure the association between wealth and health operating at the individual or household level. However, we believe this is an interesting and important finding that should be explored in future work, including an investigation of the role of between- and within-neighbourhood socio-economic inequalities.

'We measured the correlations between neighbourhood U5M and neighbourhood socioeconomic and living environment indicators using the non-parametric Spearman's rank method. We measured the

correlations across all GAMA neighbourhoods, and separately across urban and peri-urban neighbourhoods.’

We strongly agree with the Reviewer’s suggestion to explore socio-economic differences in child mortality across Accra. The Reviewer has also raised the same concern that we have with regard to reporting child mortality by neighbourhood quintile of SES. Our concern is as follows: wealthier neighbourhoods on average had higher under-five mortality (reported in the manuscript in Table 4 under the ‘all’ column and we illustrate in Fig. 3 below), and also tended to be located in the more densely populated urban core (see Fig. 3 below). It is likely therefore that the median household consumption masks within-neighbourhood socio-economic inequality and the high under-five mortality reflects the reality of the poorest and most vulnerable groups rather than truly higher under-five mortality among more affluent groups. Notably though we discuss pathways by which inequality can negatively impact health for all residents (lines) This is discussed further in Responses #11 and #16.

To overcome this limitation of within-unit inequality, as noted by the Reviewer, it is necessary to estimate under-five mortality by quintiles of household SES rather than neighbourhood SES. This requires substantial further analysis and is the aim of a current manuscript in progress, in which we are assessing the SES-child mortality relationship at the household level for Accra and all of Ghana’s 240 districts.

Fig. 3. Neighbourhood under-five mortality rates in 2010 by quintile of neighbourhood socio-economic status (measured as median household consumption).

16. As it is mentioned, using neighbourhoods could lead to mistakes at the analysis, not finding differences when there are census units with high U5 mortality within the neighbourhood or vice versa. The authors should use census units and this sentence should make you seriously consider it: “A general pattern of higher child mortality extended across inner city and more industrial neighbourhoods despite higher median household consumption and generally higher levels of post-primary education among women in these neighbourhoods relative to their peri-urban counterparts. Moreover, child mortality tended to be higher in better-off neighbourhoods of urban GAMA.”

We were unfortunately unable to do our spatial analysis at the enumeration area (EA) level. At such a fine spatial scale, the numbers of births (and women) in each five-year age-group of women were too small to robustly estimate under-five mortality rates. 39% of EAs across all age groups had less than 25 births recorded; this was reduced to 7% at the neighbourhood level. As discussed in Response 11, we updated the manuscript text to acknowledge that our findings, as with all spatial estimation, may be sensitive to the geographic unit used.

While we recognize the potential influence of the Modifiable Areal Unit Problem, we feel that our findings at the neighbourhood level are particularly relevant in the context of Ghana. Specifically, they are defined by the Ghana Statistical Service and are recognised as the administrative unit above EA, at which the population threshold to classify areas as urban versus rural is applied. Moreover, there is substantial evidence that supports the important influence of local neighbourhood factors on individual disease risk, exposure and susceptibility, and, hence, population health inequalities. We edited the manuscript to better explain the motivation for our choice of spatial unit (p.6 and p.9, respectively):

‘There is substantial evidence that health outcomes vary at small spatial scales^{3 31-33} and local neighbourhood factors are increasingly recognised as important drivers of population health inequalities.^{2 3 21-23}’

'We obtained a shapefile from the Ghana Statistical Service with all GAMA EAs, localities and districts geocoded according to the 2010 census geographies. Localities within GAMA were the neighbourhood units used in our analysis, each containing between one and 95 EAs. We linked the census data and the shapefile using codes that uniquely identified EAs to determine each individual's neighbourhood of residence. Neighbourhoods were defined by the Ghana Statistical Service and are the administrative units at which urban versus rural classification is defined in Ghana; those with 5,000 inhabitant or more are considered urban, and rural otherwise.'

This relevance is further supported by the high level of heterogeneity in child mortality that we found at the neighbourhood level, described below (pp. 13-14). The neighbourhoods are sufficiently small to examine spatial variability in under-five mortality at a high resolution.

'Across all of GAMA, neighbourhood U5M varied almost five-fold, ranging from 28 deaths per 1,000 live births (95% CrI 8-63) in the Fantsenkor neighbourhood located in the north of GAMA to 138 deaths per 1,000 live births (95% CrI 111-167) in the Mamobi neighbourhood in the city's urban core (Figure 2a and Figure 3). The variation was higher across peri-urban neighbourhoods compared to across urban neighbourhoods, ranging from 28 to 116 versus from 52 to 137 deaths per 1,000 live births.'

In recognition of the potential influence of our choice of zoning scheme, we are careful to avoid causal interpretation of the results from our correlation analyses. Although we discuss ways that within neighbourhood inequalities could contribute to worse outcomes for residents across the socio-economic spectrum, we added the following to the text to explicitly caution against over-interpretation of the results (p. 16):

'It is important to highlight, however, that we cannot draw firm conclusions regarding the harmful impact of localised socio-economic inequalities on the health of individuals within neighbourhoods from these observed correlations in our study. Associations at the area level that include neighbourhoods with diverse populations may simply obscure any association between wealth and health operating at the individual or household level.'

Similarly, where we discuss the inverse association between child mortality and improved neighbourhood living and socio-economic conditions in peri-urban areas, we have added explicit warning against causal interpretation (p. 18):

'Our finding of lower U5M in neighbourhoods with improved socio-economic and living conditions in the peri-urban area is consistent with evidence of the influence of the neighbourhood environment on population health.^{2 3 21-23} It may reflect greater access to services including health care for women in these neighbourhoods, which can influence the health of mothers and their children. However, these reported ecological correlations only provide broader context to the observed spatial patterns of U5M across GAMA and should not be interpreted as causal.'

17. This study needs to do an estimation of neonatal mortality because it could explain considerably the differences founded.

Although deaths in the first year account for an increasing proportion of deaths under five years of age globally, child deaths (in ages one to four years) remain high given the levels of infant mortality in West Africa.⁴⁹ Our outcome of under-five mortality is therefore particularly relevant given the context. We agree with the reviewer that further knowledge of the age pattern of deaths under-five and how this varies across neighbourhoods could better inform prevention efforts. Unfortunately, the available demographic methods for census data do not reliably estimate neonatal or infant mortality from summary birth history data. We acknowledge this limitation in the discussion (p.20):

'We also acknowledge that intervention efforts to prevent child mortality would benefit from more detailed information on the underlying causes of these deaths and age at which GAMA children are most vulnerable.^{50 51} In our study, we were unable to separately estimate neonatal, infant, or cause-specific mortality rates with the available methods for mortality estimation from summary birth history data and are unaware of any other data that is representative at the neighbourhood level and would enable this analysis'

18. It also should do an estimation of U5 mortality of the non-censused population (illegal immigrants) and how not including them, could affect the results.

To address this comment, we now clarify in the manuscript that our access to the full microdata of the census included records on all people present in GAMA at the time of enumeration (pp. 18-19 and included below). The data analysed were from the entire population of women aged 25-49 and included those living in regular housing (n=696,279), homeless women/outdoor sleepers (n=13,950), women who were in schools, hospitals, army and service barracks and prisons (n=2,361) and a small proportion from other locations (n=991). People were not excluded from the census or this analysis based on migration status.

'Our use of census data also enabled straightforward linkage of mortality outcomes with data on socio-economic and living conditions at a common, local spatial scale and avoided issues of sample representativeness. Specifically, the data analysed were of all women aged 25-49 present in GAMA at the time of enumeration, and included those living in regular housing (n=696,279), homeless women/outdoor sleepers (n=13,950), women who were in schools, hospitals, army and service barracks and prisons (n=2,361) and a small proportion from other locations (n=991). People were not excluded from the census or this analysis based on migration status.'

19. Also, the access to obstetrics care and maternal health need to be assessed

Assessment of access to obstetrics and maternal healthcare was outside the scope of the current analysis, largely because the data were not available in the census, which was our data source for contextual factors related to individuals and households within neighbourhoods. To our knowledge, only the Demographic and Health Surveys collect detailed data on maternal healthcare utilisation, but these data are not available at the neighbourhood level. We added text to the discussion to highlight the reviewer's point that differential access to quality care may also contribute to inequalities in mortality by impacting both maternal and child health (p.18):

'It may also reflect greater access to services including health care for women in these neighbourhoods, which can influence the health of mothers and their children.'

20. Using U5 mean to compare neighbourhoods could cause also the same mistakes described above. The authors could classify neighbourhoods by decils or quintiles, depending on socioeconomic variables, and compare results.

Please see our response to Comment #15. Quantifying under-five mortality by different measures of socio-economic level and inequality was outside the scope of this spatial analysis, which is focused on describing spatial patterns of U5M and their relationship between neighbourhood characteristics, included socio-economic level. We agree that the socioeconomic variation in U5M is of great interest, and it is currently the subject of a paper in progress that will include Accra and all of Ghana.

21. Lastly, the conclusion of this investigation give more importance to the health care services than to the social determinants of health, which, in my opinion, it merits a deeper reflection.

Please see Responses #4 and #10 where we added text on the social determinants of health. In addition, we re-wrote the conclusion to better emphasize the role of social determinants of health and frame health services as an opportunity to mitigate inequalities that arise owing to children being born into different socio-economic and living conditions (pp. 21-22):

'The determinants of child mortality are multi-faceted, operate at multiple levels and can also interact with one another. As Ghana continues to urbanise, its cities and metropolitan areas, including GAMA, will play an increasingly important role in building on national child survival efforts. The heterogeneity in child mortality across Accra's neighbourhoods highlights the need for an explicit focus on equity in the context of rapid urbanisation in SSA, with an emphasis on the social determinants of health. Concentrated deprivation in households and neighbourhoods can compound the risk of child mortality. Our study identified neighbourhoods with high child mortality in GAMA's central and industrial areas, where the urban poor may still face financial and physical barriers to accessing health services,^{41 52} potentially compounding health risks associated with disadvantaged social and living conditions. Universal access to high quality healthcare services can mitigate mortality inequalities in settings where children are born into different socio-economic and environmental circumstances.^{50 53 54} There are proven, scalable healthcare interventions that reduce under-five mortality, the causes of which are often preventable and/or treatable.^{50 55} In many LMICs the rise in facility births has not produced the expected improvements in child mortality, demonstrating the importance of continued access to quality care throughout the early years of life for all children and mothers across the socio-economic spectrum.^{53 56}

Child mortality in peri-urban neighbourhoods was on average lower than inner-city and industrial areas, but it was also more variable and inversely correlated with characteristics that indicate improved socio-economic and living environments of neighbourhoods. Complementary investment in developing infrastructure and services in neighbourhoods outside of the urban core, while ensuring that conditions in densely populated central and industrial areas do not deteriorate, could further contribute to improving child mortality and promoting health equity.'

Editor's Comments

We followed the instructions to edit the manuscript title and statement on ethical approval, and completed the STROBE checklist for observational studies. We would be happy to further modify as needed to meet the journal's requirements.

References

1. Ghana Statistical Service (GSS). Ghana Population and Housing Census. In: Service GS, ed. Accra, Ghana, 2010.
2. Borrell LN, Diez Roux AV, Rose K, et al. Neighbourhood characteristics and mortality in the Atherosclerosis Risk in Communities Study. *Int J Epidemiol* 2004;33(2):398-407. doi: 10.1093/ije/dyh063
3. Boing AF, Boing AC, Cordes J, et al. Quantifying and explaining variation in life expectancy at census tract, county, and state levels in the United States. *Proc Natl Acad Sci* 2020;117(30):17688. doi: 10.1073/pnas.2003719117

4. Nakaya T, Honjo K, Hanibuchi T, et al. Associations of all-cause mortality with census-based neighbourhood deprivation and population density in Japan: a multilevel survival analysis. *PLoS One* 2014;9(6):e97802. doi: 10.1371/journal.pone.0097802
5. Owusa G. Decentralized development planning and fragmentation of metropolitan regions: the case of the Greater Accra Metropolitan Area, Ghana. *Ghana Journal of Geography* 2015;7:1-24.
6. Yankson P, Bertrand M. Challenges of urbanization in Ghana. In: Ardayfio-Schandorf E, ed. *The mobile city of Accra: urban families, housing and residential places* 2012:25-46.
7. Agyei-Mensah S, Owusu G. Segregated by neighbourhoods? A portrait of ethnic diversity in the neighbourhoods of the Accra Metropolitan Area, Ghana. *Popul Space Place* 2010;16(6):499-516.
8. Akubia J, Bruns A. Unravelling the frontiers of urban growth: Spatio-temporal dynamics of land-use change and urban expansion in Greater Accra Metropolitan Area, Ghana. *Land* 2019;8(9):131.
9. Commission on Social Determinants of Health. *Closing the gap in a generation: health equity through action on the social determinants of health: final report of the commission on social determinants of health*: World Health Organization 2008.
10. Marmot M. Social determinants of health inequalities. *Lancet* 2005;365(9464):1099-104.
11. Chao F, You D, Pedersen J, et al. National and regional under-5 mortality rate by economic status for low-income and middle-income countries: a systematic assessment. *Lancet Glob Health* 2018;6(5):e535-e47. doi: 10.1016/S2214-109X(18)30059-7
12. Ezzati M, Webster CJ, Doyle YG, et al. Cities for global health. *Bmj* 2018;363:k3794. doi: 10.1136/bmj.k3794 [published Online First: 2018/10/05]
13. Montgomery M, Hewett PC. Urban poverty and health in developing countries: household and neighborhood effects. *Demography* 2005;42(3):397-425. doi: 10.1353/dem.2005.0020
14. Fink G, Günther I, Hill K. Slum residence and child health in developing countries. *Demography* 2014;51(4):1175-97. doi: 10.1007/s13524-014-0302-0 [published Online First: 2014/06/05]
15. Günther I, Harttgen K. Deadly cities? Spatial inequalities in mortality in sub-Saharan Africa. *Popul Dev Rev* 2012;38(3):469-86. doi: 10.1111/j.1728-4457.2012.00512.x
16. Kyu HH, Shannon HS, Georgiades K, et al. Association of urban slum residency with infant mortality and child stunting in low and middle income countries. *Biomed Res Int* 2013;2013:604974. doi: 10.1155/2013/604974
17. Timæus IM, Lush L. Intra-urban differentials in child health. *Health Transit Rev* 1995;5(2):163-90.
18. Musterd S. *Handbook of urban segregation*: Edward Elgar Publishing 2020.
19. Elliott P, Wartenberg D. Spatial epidemiology: current approaches and future challenges. *Environmental health perspectives* 2004;112(9):998-1006. doi: 10.1289/ehp.6735 [published Online First: 2004/06/17]
20. Weeks JR, Hill AG, Getis A, et al. Ethnic residential patterns as predictors of intra-urban child mortality inequality in Accra, Ghana. *Urban Geogr* 2006;27(6):526-48. doi: 10.2747/0272-3638.27.6.526 [published Online First: 2006/01/01]
21. Diez Roux AV. Neighborhoods and health: What do we know? What should we do? . *Am J Public Health* 2016;106(3):430-31. doi: 10.2105/ajph.2016.303064
22. Parks MJ, Dodoo FN-A, Ayernor PK. Applying neighborhood-effect research to a global south city: A case study of collective efficacy in Accra, Ghana's low-income areas. *The Global South* 2014;8(2):119-38.
23. Pearce N. The ecological fallacy strikes back. *J Epidemiol Community Health* 2000;54(5):326. doi: 10.1136/jech.54.5.326
24. Verhulst A. Child mortality estimation: An assessment of summary birth history methods using microsimulation. *Demogr Res* 2016;34:1075-128. doi: 10.4054/DemRes.2016.34.39
25. Rajaratnam JK, Tran LN, Lopez AD, et al. Measuring under-five mortality: validation of new low-cost methods. *PLoS Med* 2010;7(4):e1000253. doi: 10.1371/journal.pmed.1000253
26. Hill K, You D, Inoue M, et al. Child mortality estimation: Accelerated progress in reducing global child mortality, 1990-2010. *PLoS Med* 2012;9(8):e1001303. doi: 10.1371/journal.pmed.1001303

27. Jorgenson AK, Rice J. Urban slum growth and human health: a panel study of infant and child mortality in less-developed countries, 1990–2005. *J Poverty* 2010;14(4):382-402. doi: 10.1080/10875549.2010.517073
28. Rice J, Rice JS. The concentration of disadvantage and the rise of an urban penalty: urban slum prevalence and the social production of health inequalities in the developing countries. *Int J Health Serv* 2009;39(4):749-70. doi: 10.2190/HS.39.4.i [published Online First: 2009/11/26]
29. Ezeh A, Oyebode O, Satterthwaite D, et al. The history, geography, and sociology of slums and the health problems of people who live in slums. *Lancet* 2017;389(10068):547-58. doi: 10.1016/S0140-6736(16)31650-6
30. Kimani-Murage EW, Fotso JC, Egondi T, et al. Trends in childhood mortality in Kenya: The urban advantage has seemingly been wiped out. *Health Place* 2014;29:95-103. doi: <https://doi.org/10.1016/j.healthplace.2014.06.003>
31. Dwyer-Lindgren L, Squires ER, Teeple S, et al. Small area estimation of under-5 mortality in Bangladesh, Cameroon, Chad, Mozambique, Uganda, and Zambia using spatially misaligned data. *Popul Health Metr* 2018;16(1):13. doi: 10.1186/s12963-018-0171-7
32. Dwyer-Lindgren L, Stubbs RW, Bertozzi-Villa A, et al. Variation in life expectancy and mortality by cause among neighbourhoods in King County, WA, USA, 1990–2014: a census tract-level analysis for the Global Burden of Disease Study 2015. *Lancet Public Health* 2017;2(9):e400-e10. doi: [https://doi.org/10.1016/S2468-2667\(17\)30165-2](https://doi.org/10.1016/S2468-2667(17)30165-2)
33. Rashid T, Bennett JE, Paciorek CJ, et al. Life expectancy and risk of death in 6791 communities in England from 2002 to 2019: high-resolution spatiotemporal analysis of civil registration data. *Lancet Public Health* 2021;6(11):e805-e16. doi: 10.1016/S2468-2667(21)00205-X
34. Monnat SM, Peters DJ, Berg MT, et al. Using census data to understand county-level differences in overall drug mortality and opioid-related mortality by opioid type. *Am J Public Health* 2019;109(8):1084-91. doi: 10.2105/AJPH.2019.305136
35. Kwan M-P. The limits of the neighborhood effect: Contextual uncertainties in geographic, environmental health, and social science research. *Ann Am Assoc Geogr* 2018;108(6):1482-90. doi: 10.1080/24694452.2018.1453777
36. Arku RE, Bennett JE, Castro MC, et al. Geographical inequalities and social and environmental risk factors for under-five mortality in Ghana in 2000 and 2010: Bayesian spatial analysis of census data. *PLoS Med* 2016;13(6):e1002038. doi: 10.1371/journal.pmed.1002038
37. Burstein R, Henry NJ, Collison ML, et al. Mapping 123 million neonatal, infant and child deaths between 2000 and 2017. *Nature* 2019;574(7778):353-58. doi: 10.1038/s41586-019-1545-0
38. United Nations Inter-agency Group for Child Mortality Estimation. *Levels & Trends in Child Mortality*. New York: UN, 2020.
39. Kayode GA, Grobbee DE, Koduah A, et al. Temporal trends in childhood mortality in Ghana: Impacts and challenges of health policies and programs. *Glob Health Action* 2016;9:31907-07. doi: 10.3402/gha.v9.31907
40. Ghana National Health Insurance Act (Act 650), 2003.
41. Blanchet NJ, Fink G, Osei-Akoto I. The effect of Ghana's National Health Insurance Scheme on health care utilisation. *Ghana Med J* 2012;46(2):76-84.
42. Ministry of Health Ghana. *Guidelines for implementing the exemption policy on maternal deliveries*. Accra, Ghana: Ministry of Health, Ghana, 2004.
43. Ministry of Health Ghana. *Under five's child health policy: 2007-2015*. Accra, Ghana: Ministry of Health, Ghana, 2007.
44. Witter S, Garshong B, Ridde V. An exploratory study of the policy process and early implementation of the free NHIS coverage for pregnant women in Ghana. *Int J Equity Health* 2013;12(1):16. doi: 10.1186/1475-9276-12-16
45. Ghana Statistical Service (GSS), Ghana Health Service (GHS), ICF International. *Ghana Demographic and Health Survey 2014*. Rockville, Maryland, USA: GSS, GHS, and ICF International, 2015.

46. Deaton A. The analysis of household surveys: a microeconomic approach to development policy. Baltimore, MD: Published for the World Bank [by] Johns Hopkins University Press, 1997.
47. Deaton A, Zaidi S. Guidelines for constructing consumption aggregates for welfare analysis. LSMS Working Paper No. 135. Washington, D.C.: World Bank 2002.
48. Cavanaugh A, Bixby H, Schmidt AM, et al. Locating poverty and inequality: an application of small area estimation methods using survey and census data from Ghana. Manuscript in preparation 2021
49. Eilerts H, Romero Prieto J, Eaton JW, et al. Age patterns of under-5 mortality in sub-Saharan Africa during 1990–2018: A comparison of estimates from demographic surveillance with full birth histories and the historic record. *Demographic Research* 2021;44:415-42. doi: 10.4054/DemRes.2021.44.18
50. Bhutta ZA, Das JK, Bahl R, et al. Can available interventions end preventable deaths in mothers, newborn babies, and stillbirths, and at what cost? *Lancet* 2014;384(9940):347-70. doi: 10.1016/s0140-6736(14)60792-3
51. Keats EC, Das JK, Salam RA, et al. Effective interventions to address maternal and child malnutrition: An update of the evidence. *Lancet Child Adolesc Health* 2021 doi: 10.1016/s2352-4642(20)30274-1
52. Earth Institute Millennium Cities Initiative. AMA community upgrading profile: Nima-Maamobi drain area. New York: The Earth Institute at Columbia University, 2012.
53. Roder-DeWan S, Nimako K, Twum-Danso NAY, et al. Health system redesign for maternal and newborn survival: rethinking care models to close the global equity gap. *BMJ Glob Health* 2020;5(10):e002539. doi: 10.1136/bmjgh-2020-002539 [published Online First: 2020/10/16]
54. Kruk ME, Gage AD, Joseph NT, et al. Mortality due to low-quality health systems in the universal health coverage era: A systematic analysis of amenable deaths in 137 countries. *Lancet* 2018;392(10160):2203-12. doi: 10.1016/s0140-6736(18)31668-4
55. Bhutta ZA, Das JK, Walker N, et al. Interventions to address deaths from childhood pneumonia and diarrhoea equitably: what works and at what cost? *Lancet* 2013;381(9875):1417-29. doi: 10.1016/S0140-6736(13)60648-0
56. Campbell OMR, Calvert C, Testa A, et al. The scale, scope, coverage, and capability of childbirth care. *Lancet* 2016;388(10056):2193-208. doi: 10.1016/s0140-6736(16)31528-8